# Investigating the Relationship between Obstructive Sleep Apnoea, Inflammation and Cardio-Metabolic Diseases

**DOI:** 10.3390/ijms24076807

**Published:** 2023-04-06

**Authors:** Abdulmohsen Alterki, Mohamed Abu-Farha, Eman Al Shawaf, Fahd Al-Mulla, Jehad Abubaker

**Affiliations:** 1Department of Otolaryngology Head & Neck Surgery, Zain and Al Sabah Hospitals and Dasman Diabetes Institute, Dasman 15462, Kuwait; 2Department of Biochemistry and Molecular Biology, Dasman Diabetes Institute, Dasman 15462, Kuwait; 3Department of Genetics and Bioinformatics, Dasman Diabetes Institute, Dasman 15462, Kuwait

**Keywords:** obstructive sleep apnoea, inflammation, cardiovascular disease, CPAP, reactive oxygen species, hypoxia-inducible factor-1

## Abstract

Obstructive sleep apnoea (OSA) is a prevalent underdiagnosed disorder whose incidence increases with age and weight. Uniquely characterised by frequent breathing interruptions during sleep—known as intermittent hypoxia (IH)—OSA disrupts the circadian rhythm. Patients with OSA have repeated episodes of hypoxia and reoxygenation, leading to systemic consequences. OSA consequences range from apparent symptoms like excessive daytime sleepiness, neurocognitive deterioration and decreased quality of life to pathological complications characterised by elevated biomarkers linked to endocrine-metabolic and cardiovascular changes. OSA is a well-recognized risk factor for cardiovascular and cerebrovascular diseases. Furthermore, OSA is linked to other conditions that worsen cardiovascular outcomes, such as obesity. The relationship between OSA and obesity is complex and reciprocal, involving interaction between biological and lifestyle factors. The pathogenesis of both OSA and obesity involve oxidative stress, inflammation and metabolic dysregulation. The current medical practice uses continuous positive airway pressure (CPAP) as the gold standard tool to manage OSA. It has been shown to improve symptoms and cardiac function, reduce cardiovascular risk and normalise biomarkers. Nonetheless, a full understanding of the factors involved in the deleterious effects of OSA and the best methods to eliminate their occurrence are still poorly understood. In this review, we present the factors and evidence linking OSA to increased risk of cardiovascular conditions.

## 1. Introduction

Impacting approximately 25% of men and 13% of women, and with a prevalence of 56% worldwide [1,2,3], obstructive sleep apnoea (OSA) is characterised by the partial or complete collapse of the upper airway, preventing normal ventilation during sleep. As a result, individuals with OSA often feel fatigued and suffer from daytime sleepiness, lack of focus, reduced vigilance, memory impairment and overall reduced quality of life (QoL). Although male sex, advanced age and obesity are known to increase the likelihood of developing OSA, other variables such as race/ethnicity, family history and craniofacial dysmorphisms also play a role in developing OSA [4]. Additional signs and symptoms of OSA include snoring, choking and gasping during sleep, nocturia, insomnia and morning headaches [5,6]. Untreated OSA results in frequent arousal and desaturation during sleep. As a result, OSA leads to fragmented sleep and recurrent episodes of intermittent hypoxia (IH) and reoxygenation. Long-lasting hypoxemia and the resulting hypoxia can have severe health consequences, increasing the risk of developing cardiovascular diseases, including systemic hypertension, atherosclerosis, coronary artery disease, stroke and metabolic conditions such as insulin resistance and type 2 diabetes (T2D) [7,8]. In addition, OSA is associated with cognitive impairment, dementia, depression and motor vehicle accidents due to reduced daytime alertness [9,10]. Obesity is the primary risk factor that can be altered to reduce the likelihood of developing OSA; even modest weight control was successful in reducing the occurrence of sleep-disordered breathing, according to a population-based cohort analysis of 690 subjects. The study found that for every 10% increase in body weight, the apnoea-hypopnoea index (AHI) increased by nearly 32% [11].

There is an association between OSA and expanding waistlines and necks, thus OSA risk factors include a neck circumference greater than 17 inches in males and 16 inches in women. After controlling for BMI, neck circumference still appears to be a predictor of OSA, and it may even provide a stronger correlation with specific measures of disease severity in patients with OSA. Smoking, a family history of OSA and nocturnal nasal congestion are additional, less well-established risk factors. Among the various treatment methods for OSA, continuous positive airway pressure (CPAP) is the first-line treatment for patients with OSA and excessive daytime sleepiness due to its high efficacy rates [12,13]. Furthermore, CPAP is one of the best management options for relieving symptoms of classic OSA and improving the QoL of patients [14]. Additionally, the use of CPAP was associated with improvements in cognitive ability [15,16], insulin resistance [17,18] and reductions in blood pressure in patients with OSA and resistant hypertension [19,20]. On the other hand, a recent meta-analysis showed that the use of CPAP in patients with OSA was neither associated with reduced cardiovascular outcomes and death nor improved glycemic control in patients with type 2 diabetes [21,22,23,24].

OSA is associated with an increased risk of cardiovascular disease [25] and metabolic complications; current views explain these associations with increased sympathetic vasoconstrictor activity, oxidative stress and inflammation triggered by OSA-associated IH. Here, we review the most recent evidence of the pathophysiological consequences of severe OSA, focusing on the role of inflammation in the development of cardiovascular complications.

## 2. OSA and Inflammation

The continuous stress caused by recurrent snoring and repetitive closure and re-opening of the upper airway in patients with OSA can lead to mucosa inflammation [26,27]. Low-grade inflammation causes remodelling of the upper airway, with increased deposition of connective tissue in the mucosa and muscles. Consequently, introducing changes to the morphology of palatopharyngeal muscle, inflammation and denervation contribute to upper airway obstruction during sleep in patients with OSA [28,29]. Therefore, systemic and localised inflammation induced by IH and the repeated cycles of hypoxia and reoxygenation (Figure 1) features several events: induction of oxidative stress, reactive oxygen species (ROS) production [30], activation of a critical proinflammatory transcription factor (nuclear factor κB (NF-kB)) [31], elevated expression of proinflammatory cytokines and chemokines [32,33], recruitment and infiltration of the proinflammatory M1 macrophages [30,34] in different tissues such as vessels, heart, adipose tissue and liver. Eventually, this contributes to vascular remodelling, metabolic dysfunction [35] and atherosclerosis [36].

Indeed, upper airway tissue biopsies from patients with OSA showed subepithelial oedema and increased inflammatory cell infiltration [28,37,38,39]. In addition, some studies have also reported increased levels of exhaled nitric oxide in patients with OSA [40,41,42,43]. However, a recent meta-analysis showed no significant differences in exhaled nitric oxide relative to control groups [44]. Although the association between localised inflammation and OSA is well established, whether systemic inflammation correlates with OSA is debatable. Some studies reported elevated levels of biomarkers of systemic inflammation (Table 1), including proinflammatory cytokines such as interleukin (IL)-6 and C-reactive protein (CRP) in patients with OSA [45,46]. Contrariwise, others found no association between OSA and CRP or IL-6 levels [47,48]. In an Icelandic cohort study, OSA severity affected IL-6 levels only in obese subjects, while CRP levels were solely associated with OSA severity in obese men and postmenopausal women [49]. This well-established correlation between obesity, a common OSA risk factor, and levels of IL-6 and CRP makes such molecules potential confounding factors in these studies. More recently, a study proposed Pentraxin-3 (PTX-3) as a biomarker of inflammation in patients with OSA as well as an indicator of disease severity [50].

Studies suggest that OSA-associated systemic inflammation results from the overflow of inflammatory cytokines, which reach the bloodstream from the upper airway mucosa. This mechanism has been proposed in other respiratory conditions such as chronic obstructive pulmonary disease [51]. Vincente et al. explored the possible association between local upper airway and systemic inflammation by assessing the levels of proinflammatory markers—IL-6, IL-8, tumour necrosis factor α (TNF-α) and CRP—and activated leukocytes in both pharyngeal lavage (PHAL) and plasma samples of patients with severe OSA, snorers and healthy control subjects [39]. Though higher levels of IL-6, IL-8 and CD4^+^ T-cells were found in the PHAL of severe OSA patients (apnoea–hypopnoea index (AHI) ≥ 30 per h), plasma inflammatory cytokines did not differ between patients with severe OSA and control groups. Thus, these findings do not support a link between local and systemic inflammation in patients with OSA [39].

Recent views support a strong link between OSA and systemic inflammation [52]. Two meta-analyses evaluating the association between CRP and other inflammatory biomarkers (Table 1), i.e., IL-6, IL-8 and TNF-α and adhesion molecules such as Intercellular Adhesion Molecule (ICAM), Vascular Cell Adhesion Molecule (VCAM) and selectins reported a positive association with OSA [53,54,55]. Moreover, an elevation of cytokines and adhesion molecules correlated with the severity of the nocturnal hypoxaemia [56]. The link between OSA and systemic inflammation was demonstrated in a recent publication reporting inflammasome activation in people with OSA [57]. In a group of people with OSA, there was an increase in the level of an inflammasome, nucleotide-binding oligomerization domain-like receptor (NLR) family pyrin domain containing 3 (NLRP3), which is an intracellular multiprotein complex involved in regulating the secretion and pyroptosis of IL-1β and is implicated in the pathogenesis of sterile inflammatory diseases. The study showed that under conditions of intermittent hypoxia, monocytes increased NLRP3 signalling in a HIF1α-dependent manner [57]. In line with this, anti-inflammatory therapy in patients with OSA leads to a modest improvement in OSA severity in adults and children [58]. Nonetheless, the effect of CPAP treatment on systemic inflammation biomarkers has been inconsistent [59,60,61,62,63], whereby only some studies demonstrated a positive effect of CPAP [56]. Furthermore, systemic inflammation has been implicated in impaired respiratory chemoreflexes and plasticity, thus undermining ventilatory control [64]. However, recent studies have shown inflammation and IH’s positive impact on respiratory plasticity [65,66].

**Table 1 ijms-24-06807-t001:** List of elevated biomarkers reported in studies involving patients with OSA.

Condition	Biomarker ↑	Reference
**OSA-induced systemic inflammation biomarkers**	IL-6	[39,46,53,55]
	IL-8	[39,55]
	PTX-3	[50]
	CRP	[45,54]
	TNF-α	[39]
	ICAM	[53,55,56]
	VCAM	[53,55,56]
**OSA and obesity biomarkers**	IL-1β	[67]
	IL-6	[67,68]
	IL-17	[69,70]
	IL-23	[69,70]
	TNF-α	[67,68]
	(MCP)-1	[67]
	resistin	[67]
	leptin	[67,70,71,72]

## 3. OSA and Obesity

OSA is intimately associated with obesity, with OSA being diagnosed in more than 30% of people with obesity and 50–98% of people with morbid obesity [73]; thus, a higher BMI score increases the likelihood of developing OSA [74]. Both conditions are associated with severe cardio-metabolic complications. The rising prevalence of OSA in the past 20 years concurred with increasing prevalence of obesity worldwide [75]. Obesity is an established major risk factor for OSA [74,76]. Three extensive cohort studies have shown that weight gain is associated with an increased risk of developing moderate to severe OSA and correlated with increased AHI. In contrast, weight loss coalesced with reduced AHI and OSA severity [11,77,78]. Several reports proposed obesity as a factor contributing to the pathogenesis of OSA by increasing fat deposition in the parapharyngeal fat pads and tongue, thus narrowing the upper airway [79,80]. Furthermore, obesity can cause chest and abdominal wall compression, reducing tracheal tension and contributing to a more collapsible upper airway [76,81]. Obesity was associated with the severity of blood oxygen desaturation during apnoea and hypopnoea, thus likely aggravating OSA symptoms in these patients [82].

On the other hand, the relationship between OSA and obesity may be reciprocal, as OSA may also contribute to weight gain. Many patients have reported gaining weight on the onset of OSA diagnosis [83,84]. Several aspects of OSA may contribute to the development or aggravation of obesity: the fragmented sleep pattern associated with OSA affects dietary habits and sleep duration, leading to hormonal changes that affect satiety and energy expenditure [85]. Increased daytime sleepiness and fatigue also contribute to reduced physical activity, leading to weight gain and increased risk of obesity comorbidities [73]. CPAP therapy may also impact body weight in patients with OSA. Meta-analyses of randomised controlled trials showed that CPAP therapy led to a small but significant increase in body mass index [35], particularly in high-usage patients [86]. However, these analyses involved studies of relatively short duration (median of 3 months). Contrastingly, a post hoc analysis of the Sleep Apnea Cardiovascular Endpoints (SAVE) cohort showed that long-term use of CPAP (>3 years) did not have a significant impact on body weight in patients with comorbid OSA and cardiovascular disease [87].

Notably, both OSA and obesity trigger the secretion of proinflammatory factors. An increasing body of evidence supports the common pathways by which obesity and OSA-associated IH lead to inflammation and dysfunction of adipose tissue (Figure 1). However, given the similar mechanisms that trigger low-grade inflammation in obesity and OSA, discerning between OSA-specific and obesity-triggered inflammatory biomarkers remains challenging [88]. In addition, proinflammatory cells are targeted at dysfunctional adipose tissue, producing proinflammatory factors such as TNF-α, IL-6, IL-1β, monocyte chemotactic protein (MCP)-1, resistin and leptin (Table 1) [67]. Levels of the proinflammatory cytokines IL-17 and IL-23 were significantly elevated in paediatric OSA cases [69]. In a study involving children with obesity and OSA, IL-17 was significantly associated with OSA, while IL-23 levels correlated with body fat and liver enzymes. Therefore, studies proposed these proinflammatory cytokines as potential biomarkers of paediatric OSA [70]. Additionally, leptin levels significantly increased in people with sleep disorder breathing. The rise in leptin levels correlated directly with BMI Z-score, AHI [70,71,72] and fasting insulin [70]. The relationship between OSA and cytokines was corroborated by an observational study showing the diurnal variation of IL-6, IL-8 and TNF-α with the expression of symptoms in patients with mild OSA [68].

Repeated exposure to hypoxia alters gene transcription and posttranslational protein modification, influencing metabolic and cardiovascular processes. OSA and obesity both trigger hypoxia-inducible factor 1 (HIF-1) in adipocytes, ultimately stimulating the expression of downstream angiogenic proteins to increase oxygen and nutrient delivery to adipocytes. Primarily responsible for maintaining stable oxygen metabolism, HIF-1 is an attractive target that is selectively induced by chronic IH and promoted by oxidative stress. The heterodimeric complex comprises two different proteins: HIFα and HIFβ. Both belong to a family of transcription factors constitutively expressed in cells [89]. HIF-1α is oxygen-sensitive and couples with von Hippel-Lindau protein during normoxia to trigger its degradation by proteasomes [90]. The degradation of HIF-1α is inhibited under hypoxic conditions [91]. It is estimated that HIF-1 activates over 100 distinct genes [92]. Consequently, HIF-1 is a critical transcription factor that regulates numerous processes, including the metabolism and cardiovascular system [92,93]. However, many signalling pathways in which it participates remain poorly understood. Among numerous others, it stimulates genes associated with angiogenesis or glucose uptake by cells, i.e., glucose transporter 1 and glucose transporter 4 [92,94].

Few OSA human studies have examined HIF-1α [95,96,97]. OSA patients with severe nocturnal desaturations (haemoglobin oxygen saturation, sub 75%) had higher HIF-1α mRNA expression in skin biopsies compared with those with no night-time desaturations [98]. In addition, patients with OSA had increased levels of HIF-1α serum protein [99]. Lu et al. demonstrated variations in HIF-1 protein concentrations between people with severe OSA and those with moderate or mild OSA. After two months of CPAP therapy, the levels of HIF-1 resembled those in the control group [98]. Levels of HIF-1α in patients with OSA and healthy controls did not show evening-morning fluctuations, which indicates that occasional nocturnal hypoxia in OSA results in a persistent rise in HIF-1 levels [100]. However, the deleterious effects of HIF-1 activation were mainly explored in animal and cell models of OSA that exhibit elevated HIF-1α expression [101,102,103].

## 4. OSA and Cardiovascular Disease

OSA is a recognised independent risk factor for CVD and is associated with metabolic abnormalities that are closely linked to cardiovascular diseases. In accordance with this, OSA is associated with hypertension [104,105,106], which puts patients with OSA at increased risk of developing cardiovascular complications such as coronary artery disease and stroke [107]. Hypertension affects 36.5–53.6% of patients with OSA, depending on severity of the condition [108,109]. Mechanisms driving increased blood pressure in patients with OSA are well described. Obstruction of the upper airways leads to hypoxemia, hypoxia, hypercapnia and changes in intrathoracic pressure, which, combined with frequent arousal from sleep, induces sympathetic system activation, increasing both heart rate and blood pressure. These hemodynamic changes contribute to tachycardia and hypertension, which can develop into left ventricular hypertrophy and heart failure [107].

Chronic OSA is linked to oxidative stress due to the elevated synthesis of ROS brought on by the frequent episodes of IH and reoxygenation, which contribute to the development of oxidative stress. Several cross-sectional studies have examined different oxidative stress markers in patients with OSA. For example, high-sensitivity CRP, metalloproteinase 9 and copper correlate with higher AHI and lower haemoglobin oxygen saturation [110,111]. Similar results were reported with the protein disulfide reductase thioredoxin [112,113]. In addition, Malondialdehyde, a common biomarker of lipid peroxidation, has also been associated with OSA, correlating with the duration of nocturnal oxygen desaturation below 85% [114,115,116].

Furthermore, patients with OSA presented with higher levels of oxidised LDL compared with healthy controls [117,118]. Additionally, the presence of OSA concurred with lower ferric-reducing antioxidant power (FRAP) compared with controls, which negatively correlated with AHI [119,120]. On the other hand, several oxidative stress biomarkers did not correlate with OSA: 8-isoprostane, thiobarbituric acid-reactive substances, erythrocyte catalase activity, copper-zinc superoxide dismutase (SOD) and total antioxidant capacity [121,122]. While the benefit of CPAP therapy in reducing oxidative stress biomarkers in patients with OSA is debatable [56,123], antioxidant therapy is effective in reducing oxidative stress and is a potential alternative to CPAP in patients with OSA [124,125,126]. Several studies have investigated polyphenols as a treatment for OSA. Grebe et al. reported improved endothelial health through measurements of flow-mediated dilation (FMD) of the brachial artery. This study reported lower FMD measurements in a group of patients with OSA after receiving vitamin C intravenously compared with controls [127]. Additionally, vitamins C and E have been studied for their potential to mitigate oxidative stress in laboratory rodents and other animal models. Higher levels of malondialdehyde (MDA) and advanced oxidation protein products (AOPP), both markers of oxidative stress, were found in subjects who experienced intermittent hypoxia caused by tracheal obstruction. Although the levels of MDA were not affected by the administration of antioxidants, AOPP levels were reduced considerably [128]. Antioxidant properties are found in several medications commonly used to treat diseases other than OSA. N-acetylcysteine (NAC) is a well-known mucolytic drug that is used to treat acetaminophen overdose, is required for glutathione synthesis and has antioxidant properties. In a study involving patients with OSA, oral administration of NAC for 30 days caused a significant reduction in lipid peroxidation products and significantly increased glutathione levels [124]. Currently, an ongoing clinical trial (NCT05009901) is examining the effect of using a powerful oral antioxidant (alpha lipoic acid, ALA) to treat patients with OSA by improving their cardiovascular health and reducing systemic inflammation and markers of oxidative stress. Another possible mechanism for the development of hypertension and consequent cardiovascular disease in patients with OSA is the activation of the renin-angiotensin-aldosterone system (RAAS). Meta-analyses of 13 studies assessing the role of OSA on RAAS components have reported higher plasma levels of angiotensin II in patients with OSA compared with controls and that patients with hypertension and OSA had higher aldosterone plasma levels compared with the controls [129].

IH and reoxygenation may also contribute to the development of cardiovascular disease by promoting inflammation, which is strongly associated with endothelial dysfunction, atherosclerosis and coronary artery disease [130]. Numerous animal and cell-based studies have pointed to a strong link between IH and vascular and systemic inflammation development. In mice, IH-induced atherosclerotic changes occurred with increased expression of proinflammatory cytokines, chemokines and adhesion molecules, increased migration of inflammatory cells and expansion of the macrophage population in the arterial wall [31,36,131]. In addition, the aorta of mice exposed to IH had higher expression of the proinflammatory transcription factor NF-kB, although the levels of NF-kB returned to normal after recovering from normoxia [32,132]. In humans, monocytes of patients with severe OSA showed elevated NLRP3 activity compared with monocytes from control subjects and this elevation showed a direct correlation with the AHI score and other hypoxemic indices [57]. This report showed that higher NLRP3 activity triggered inflammatory cytokines, i.e., IL-1β and IL-18, via caspase-1 and increased Gasdermin D, which consequently allowed for tissue factor to be released. As a result, plasma concentrations of tissue factor were higher in patients with OSA and systemic inflammatory comorbidities compared with controls [57]. In a prospective cohort study, the activation of NF-κB coincided with endothelial dysfunction and higher levels of its downstream targets were detected in endothelial cells of patients with OSA [133]. In particular, the effects on endothelial function were reversed after four weeks of CPAP therapy [134].

## 5. OSA and Metabolic Dysfunction

### 5.1. Insulin Resistance and Glycemic Control

While cardiovascular risk is a significant concern in people with OSA, evidence suggests that 15–30% of patients with OSA also have T2D [135,136]. Sleep disturbances, in particular OSA, are now well acknowledged to play a significant role in the development of metabolic syndrome [137,138]. OSA appears to be an independent risk factor for developing T2D, insulin resistance and glucose intolerance (Figure 2) [139,140]. Thus, in patients with T2D, OSA and higher oxygen desaturation levels were associated with higher glycated haemoglobin (HbA1c) levels [141,142,143,144].

Several studies have shown an association between IH and impaired glucose metabolism. For example, rodents exposed to chronic IH for more than 30 days exhibit higher fasting glucose and insulin levels, beta cell dysfunction and insulin resistance [145,146]. In humans, OSA-associated nocturnal hypoxemia and IH have been associated with glucose intolerance and insulin resistance, possibly contributing to the development and progression of T2D [147,148,149,150]. Nevertheless, the association between IH and OSA and how this leads to glucose metabolic impairment is still unclear. The mechanisms implicated in IH-mediated dysfunctional glucose metabolism include sympathetic nervous system stimulation, oxidative stress and inflammation, which damage the liver and pancreas. In a cross-sectional study involving Koreans with OSA, the odds ratio for insulin resistance (IR) and metabolic syndrome (MetS) increased significantly in the presence of OSA [151]. The risk of OSA was significantly associated with abdominal obesity, hypertension, hypertriglyceridemia and impaired FBG in women. This reflects a strong link between OSA and metabolic disturbances such as IR and MetS, whereby elevated AHI scores are associated with IR [152]. The connection between OSA, IR and MetS is attributed to IH and the consequent sustained oxidative stress [153]. Extended exposure to IH increases oxidative stress by augmenting the oxidation of deoxyribonucleic acid, lipids and proteins [154]. Additionally, OSA-induced fragmented sleep elicits a transient activation of the cortical and sympathetic nervous system that could initiate and maintain IR or hypertension [151,155].

As a significant insulin target organ, the liver plays an essential role in glucose homeostasis and is heavily affected by IH. In animal models, sympathetic activation in response to IH is an essential mechanism affecting hepatic glucose output and insulin signal transduction [156,157]. IH also induces oxidative stress and is associated with increased lipid peroxidation in the liver [158,159]. As a potent proinflammatory trigger, IH activates NF-κB, which activates a series of downstream proinflammatory cytokines and chemokines, leading to liver inflammation, hepatic steatosis and necrosis of hepatocytes. In animal models, exposure to IH increased TNF-α, IL-1β, IL-6, HIF-1 and the C-X-C motif ligand 2 (CXCL2) in the liver, altering hepatic glucose metabolism [160,161]. In C57BL6/J mice, exposure to IH for two weeks led to higher fasting glucose levels due to increased glucose production and stimulation of gluconeogenic pathways [162].

In the pancreas, IH was associated with impaired insulin sensitivity, reduced beta cell function and increased oxidative stress [162,163,164]. Several mechanisms may be responsible for the alterations in beta cell function in response to IH. Studies in murine models have implicated IH in beta cell proliferation, mimicking the physiological response to hyperglycemia [165,166,167,168]. In rats, IH induced beta cell replication by upregulating *Reg* family genes [169]. Additionally, IH-induced apoptosis of beta cells was reduced due to suppression of the expression of apoptosis-producing Bcl-2 and the upregulation of anti-apoptotic B cell lymphoma 2 (Bcl-2)-associated X protein (Bax) [167], while IH-induced oxidative stress was shown to promote beta cell apoptosis [167]. On the other hand, IH reduced glucose-induced insulin secretion by downregulating CD38 in hamster beta cells and pancreatic rat islets [170]. In a recent study, IH was also found to disrupt beta cell function by decreasing gamma-amino butyric acid type A (GABA_A_) receptors in the membranes of pancreatic beta cells [164]. GABA_A_ is a chloride-ion channel that couples to GABA to regulate the concentration of [Cl^−^]_i_ in beta cells, which is critical for depolarizing membranes to induce insulin secretion. This gives insights into some of the potential factors linking IH insults to the development of T2D, where GABA_A_ is downregulated in response to IH, leading to reduced insulin secretion [164]. This finding was corroborated by the reported elevation of GABA levels in paediatric patients with OSA having IH during sleep [171].

In patients with OSA, several randomised controlled trials support the beneficial impact of CPAP therapy on insulin sensitivity [172,173,174,175]. In some OSA studies, patients undergoing CPAP therapy showed improved glycemic control, which was reflected by decreased HbA1c levels [176,177,178,179]. A meta-analysis of randomised controlled trials and prospective observational studies showed that CPAP therapy improved insulin sensitivity in patients with OSA and T2D without significant reductions in HbA1c levels [18,180]. Overall, people with OSA are insulin-resistant and have increased incidence of T2D regardless of their BMI [34]. On the other hand, the severity of insulin resistance is directly correlated with nocturnal hypoxia in non-obese patients with OSA [181]. Furthermore, the high frequency of metabolic issues in OSA patients is consistent with the critical link between OSA and cardiovascular disorders like arterial hypertension, coronary artery disease, arrhythmias and heart failure identified in several clinical trials [182].

### 5.2. Lipid Metabolism

In the past two decades, an increasing body of evidence has supported the prominent role of OSA in the dysregulation of lipid metabolism. The effect of OSA on changing the lipid profile is reviewed in a recent publication by Meszaros et. al., who present a comprehensive and critical overview of the current evidence linking OSA to dyslipidemia [183]. In accordance with this, patients with OSA often have abnormal lipid metabolism, including hypercholesterolemia, increased triglycerides, decreased LDL and lipid peroxidation. Hypercholesterolemia and hypertriglyceridemia have been linked to the desaturation index, which measures the intensity of nocturnal hypoxia [184]. The desaturation index is a surrogate marker for hypoxia that contributes independently to hypercholesterolemia and hypertriglyceridemia [184].

Although poorly understood, chronic IH appears to be a key determinant of OSA, inducing the progression of dyslipidemia, systemic inflammation, oxidative stress, endothelial dysfunction and atherosclerosis in both in vitro and in vivo models [184]. OSA studies showed that IH promotes the generation of sterol regulatory element binding protein-1 (SREBP-1) and stearoyl coenzyme A desaturase-1 (SCD-1) [185,186]. Additionally, it induces lipid peroxidation and HDL dysfunction, which increase total cholesterol and cause sympathetic dysfunction [187,188,189]. Episodes of IH trigger the activation of HIF-1 in the liver that induces SREBP-1 and SCD-1, which in turn enhance triglyceride and phospholipid biosynthesis, resulting in dyslipidemia and atherosclerosis in OSA [186,190]. Additionally, IH was found to act as a potent inhibitor of lipoprotein lipase (LPL) [191], with changes in serum LPL levels correlating with the desaturation index, reflecting nocturnal hypoxia [192]. The effect of chronic IH was further explored in animal OSA models, revealing an elevation in the levels of ANGPTL4 that was HIF-1α-dependent [134]. Other studies reported increased serum ANGPTL3 levels in patients with OSA and coronary artery disease compared with people with OSA only [193]. Elevated plasma levels of ANGPTL4 and ANGPTL8 were also reported in people with OSA [194].

Furthermore, oxidative stress in OSA modulates lipid activity and generates oxidised and dysfunctional lipids. As such, lipid peroxidation and the increase in levels of atherogenic oxidised LDL cholesterol and dysfunctional HDL molecules are detected in people with OSA [195]. Oxidation of LDL molecules due to oxidative stress and inflammation is one of the prominent lipid modification events in OSA. The production of small dense LDL (sdLDL) particles is one of the frequent LDL modifications that contribute to increased levels of oxidized LDL (oxLDL) molecules [196]. In addition to oxidation, the size of LDL particles appeared to be an important factor in patients with OSA and MetS; however, OSA severity did not independently contribute to LDL phenotype alterations [197]. Previous reports found sdLDL to be associated with OSA, independent of obesity, in a group of non-obese people with OSA [198]. Although oxLDL levels were increased in previous OSA studies, a recent meta-analysis demonstrates that people with OSA have elevated levels of circulating oxLDL; nonetheless, this rise was governed by age and BMI as confounding factors in people with OSA [118].

Several hypotheses have been put forth regarding the role of HDL in preventing atherogenesis; these include the destruction of lipid hydroxyperoxides to prevent oxidation of LDL phospholipids [199,200], and most importantly, the inhibition of MCP-1 [201]. The protective activities of HDL are mediated by the liver-made enzymes paraoxonases (PON-1 and PON-3) and apolipoprotein (apoA-1) [202], which appear to be dysfunctional in OSA [195]. The expression of ApoA-1 in the liver was suppressed by inflammation and oxidative stress, leading to reduced levels of HDL [203]. According to research, not only is the protective antioxidant enzyme (PON-1) decreased in OSA, it also negatively correlates with the severity of OSA as measured by the respiratory disturbance index [204]. Thus, some studies found reduced levels of circulating HDL in people with OSA [204], while others reported a higher degree of OSA-associated HDL dysfunction that concurred with a rise in oxLDL levels [205,206]. The loss of circulating HDL particles might be attributed to the increase in serum amyloid A (SAA) levels due to inflammation, which can replace ApoA-1 in HDL, leading to rapid clearance of HDL from the circulation [207]. The antioxidant and anti-inflammatory characteristics of HDL may account, at least in part, for the antiatherogenic potential of HDL and the potential role of OSA in dyslipidemia and atherosclerosis. A reliable biomarker of HDL function is the HDL cholesterol efflux capacity (CEC), and this was shown for the first time to be reduced in people with OSA, where PBMC-derived macrophages from patients with OSA showed a suppressed ability to efflux cholesterol through the ATP-binding cassette transporter 1 (ABCA1) [206]. Considering the adverse effect of chronic IH on serum lipids and its consequences, bi-level or CPAP therapy appears to reduce the effect of oxidative stress [208] and resolve the effect of IH, thus improving the lipid profile [209].

## 6. Conclusions and Future Directions

In recent years, OSA has become a serious public health concern due to its association with numerous diseases—especially diabetes and cardiovascular complications—morbidity and death. Consequently, there is an urgent need for a deeper comprehension of its fundamental mechanics. Models of OSA features and characteristics, including animal and human models, have significantly advanced our understanding of OSA and revealed that IH and recurrent arousals are central to the development of CVD via activation of a set of intermediate pathways, including sympathetic activation, oxidative stress, inflammation and metabolic dysregulation. In addition, obesity, a common comorbid disease associated with OSA, may exacerbate the negative impact of these activating variables. The synergistic effects of obesity and the degree to which it participates in the complex processes leading to pathophysiological pathways in OSA are unclear.

Therefore, mechanistic research investigating the links between pathophysiological pathways and their activating factors is essential to identifying potential novel therapy options. Important advances in our knowledge of pathophysiology and the diversity of clinical phenotypes necessitate new methods to treat individual patients, instead of the conventional emphasis on CPAP alone. Studies focused on identifying and formulating better diagnostic markers and/or methods that would facilitate early OSA diagnosis are crucial. since prevention is the best management method for OSA, identifying patients with OSA early would help to prevent and control the occurrence of the associated cardiovascular and metabolic complications.

## Figures and Tables

**Figure 1 ijms-24-06807-f001:**
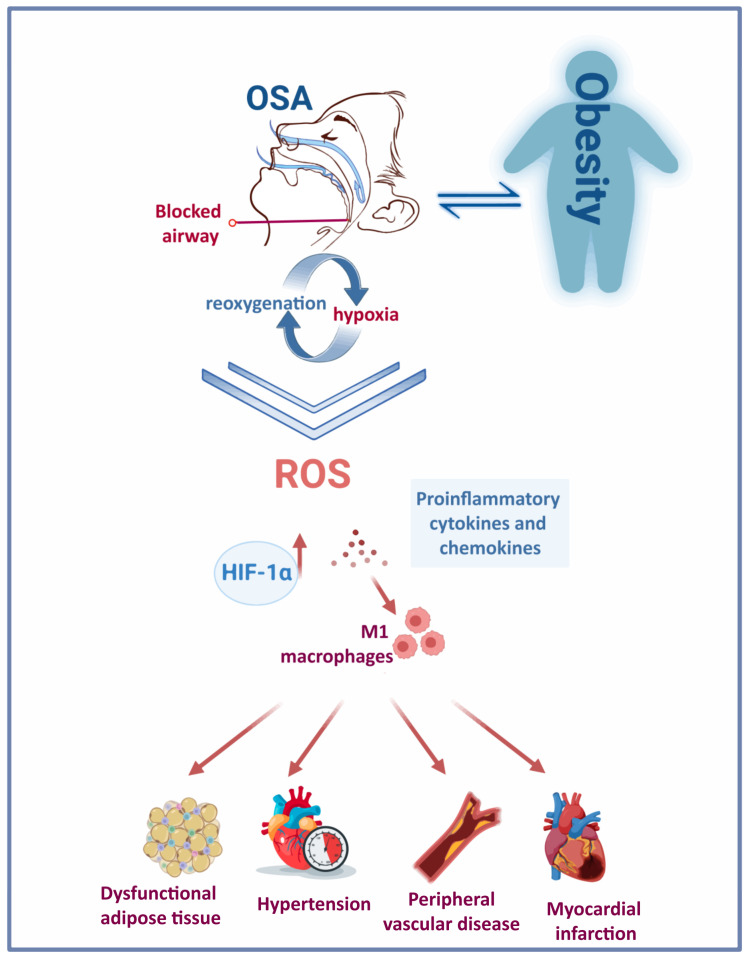
Obesity and OSA: a bidirectional interaction. The repetitive cycles of hypoxia-reoxygenation due to frequent blockage of the upper airway in people with OSA triggers the activation of various pathological mechanisms and induces a rise in HIF-1α. The chronic presence of this condition has deleterious consequences on the cardiovascular system.

**Figure 2 ijms-24-06807-f002:**
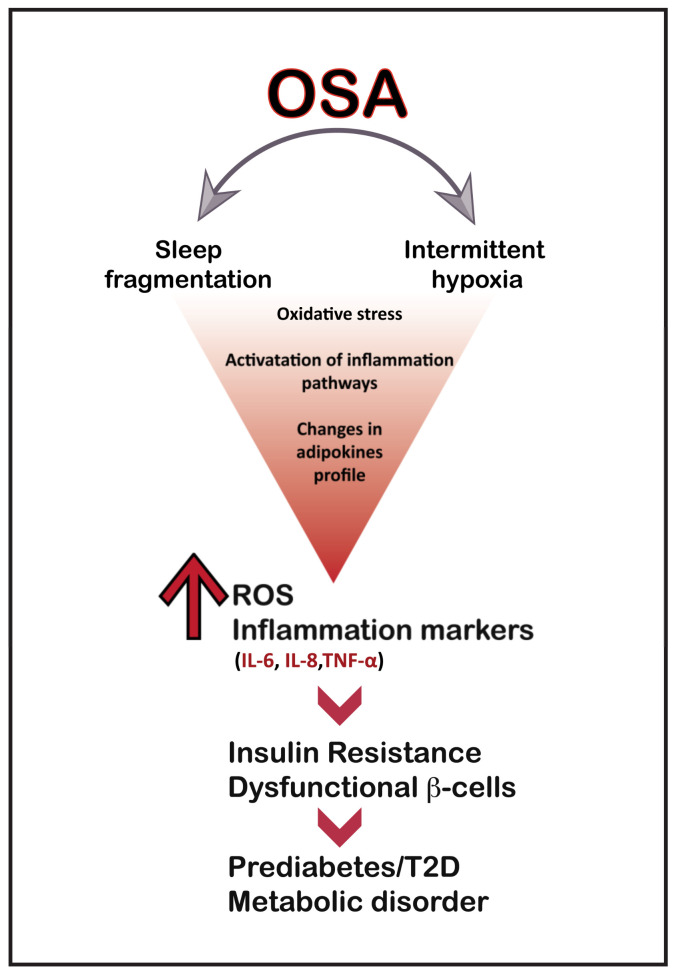
Potential mechanisms linking hypoxia to T2D.

## Data Availability

Not applicable.

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
