# Peer review of "Investigating the Relationship between Obstructive Sleep Apnoea, Inflammation and Cardio-Metabolic Diseases"

_ijms, 2023, doi:10.3390/ijms24076807_

Round 1

Reviewer 1 Report

This is a review paper presenting an extensive review of the literature about deleterious effects of OSA. The review is very well written and documented. A few remarks for improving the review::

The title should include metabolic consequences as those are mainly discussed in the review, more than cardiovascular disease. “Obstructive sleep apnea, inflammation, cardiovascular disease and metabolic consequences” would be more appropriate.

Figure 1: this diagram is not very informative, with very small font and difficult to read. A summarizing figure at the end of the paper would be more useful to provide a comprehensive picture of all the interactions between OSA, inflammation, metabolic changes, and CVD. This would improve the conclusion that doesn’t say much.

Some of the statements are repetitive for example, that OSA and IH are linked to oxidative stress, or to ROS…. These statements keep coming back

Reviewer 2 Report

The review article is moderately interesting, because despite the topic being relevant, the review is very broad.

Some topics are not well explored like “antioxidant therapy is effective in reducing oxidative stress and is a potential alternative to CPAP in patients with OSA - Line 237 - Explore more the topic, presenting concrete examples.

Line 377 – correct the reference citation “{Feingold, 2016 #226}” and the next ones.

The future perspectives could be more precise and give specific directions for further research.

Reviewer 3 Report

In this review, author described all the factors and the evidence linking Obstructive sleep apnea (OSA) to an increased risk of cardiovascular conditions in people with OSA. OSA is a well-recognized risk factor for cardiovascular and cerebrovascular diseases. The relationship between OSA and obesity is complex and reciprocal, involving interaction between biological and lifestyle factors. OSA and obesity, involve oxidative stress, inflammation, and metabolic dysregulation.  Author explained about uses of continuous positive airway pressure (CPAP) as a gold standard tool to manage OSA. It has been shown to improve symptoms and cardiac function, reduce cardiovascular risk and normalize biomarkers. It is clearly a big effort of author for doing this study, but despite of positive outcomes I have some suggestion where it can be improved.

Comments

1.     Title of the paper is Obstructive Sleep Apnea, inflammation, and Cardiovascular Disease. Author can change the title “Association of Obstructive Sleep Apnea with Cardiovascular Disease and metabolic dysfunction” or something like that.

2.     Author explained about the association of OSA with inflammation, cardiovascular disease, and metabolic dysfunction. Author can make a table having all the biomarker or cofactors affecting OSA during inflammation, obesity, cardiovascular disease and metabolic dysfunction to simplify the review if possible.

3.     Author explained about continuous positive airway pressure (CPAP) used as a treatment for patients with OSA. Author can make a separate paragraph saying, “Treatment of OSA”.

4.     Author can also make a table about all the cofactors or biomarker improved by CPAP.

5.     Author should work on introduction part. 

Round 2

Reviewer 3 Report

Author did a great effort, overall, the manuscript has been modified extensively and I am ready to accept this paper in present form.

Author Response

Many thanks for the positive comments.